# Modeling the Ignition Risk: Analysis before and after Megafire on Maule Region, Chile

Gabriela Azócar de la Cruz [1,2,3,*], Gabriela Alfaro [3,4], Claudia Alonso [2,3], Rubén Calvo [3,5] and Paz Orellana [3]

1   Department of Social Work, University of Chile, Av. Ignacio Carrera Pinto 1045, Ñuñoa, Santiago 7800284, Chile
2   Center for Climate and Resilience Research (CR), Blanco Encalada 2002, Floor 4, Santiago 8370449, Chile
3   Nucleus of Transdisciplinary Systemic Studies, University of Chile, Santiago 7820436, Chile
4   Industrial Engineering Department, University of Chile, Av. Víctor Jara 3769, Estación Central, Santiago 9170124, Chile
5   Institute of Geography, Pontificia Universidad Católica de Chile, Campus San Joaquín—Avda. Vicuña Mackenna 4860, Macul, Santiago 7820436, Chile
*   Correspondence: gazocarde@uchile.cl

**Abstract:** Wildland fires are a phenomenon of broad interest due to their relationship with climate change. The impacts of climate change are related to a greater frequency and intensity of wildland fires. In this context, megafires have become a phenomenon of particular concern. In this study, we develop a model of ignition risk. We use factors such as human activity, geographic, topographic, and land cover variables to develop a bagged decision tree model. The study area corresponds to the Maule region in Chile, a large zone with a Mediterranean climate. This area was affected by a megafire in 2017. After generating the model, we compared three interface zones, analyzing the scar and the occurrences of ignition during and after the megafire. For the construction of georeferenced data, we used the geographic information system QGIS. The results show a model with high fit goodness that can be replicated in other areas. Fewer ignitions are observed after the megafire, a high recovery of urban infrastructure, and a slow recovery of forest plantations. It is feasible to interpret that the lower number of ignitions observed in the 2019–2020 season is a consequence of the megafire scar. It is crucial to remember that the risk of ignition will increase as forest crops recover. Wildland fire management requires integrating this information into decision-making processes if we consider that the impacts of climate change persist in the area.

**Keywords:** wildfire; ignition risk; model; megafire; climate change; bagged decision tree; wildland urban interface



## 1. Introduction

Wildfires are an extreme phenomenon of great global concern [1]. The frequency and intensity of these events have progressively increased in different areas of the world [2]. Studying the causes and risks of forest fires has become a field of broad interest. Various investigations and technical reports on wildfires agree that at least 90% of these have an anthropogenic origin [3–7]. It is not entirely clear, however, how many of the fires caused by human actions are due to negligence and how many are intentional. Identifying and punishing the people responsible for wildfires is a complex problem, given that the evidence is not recordable or disappears due to the fire [8]. A series of environmental factors favor the spread of wildfires, such as the combination of temperature, humidity, and wind [4,9–11]. Added to this are research results on the influence of productive activities and the characteristics of wildland–urban interface zones [6,12,13]. All the above indicates the need to deepen the analysis of the interaction between human action and its ecological environment to investigate the conditions that affect the fire origin and spread. In addition, there is a need to connect this with the impacts of socio-environmental disasters.

Climate change is a highly complex phenomenon that has become the focus of interdisciplinary studies on the relationship between society and the environment, particularly disaster risks. Its link to wildfires exemplifies this evident relationship [14–18]. Due to climate change, different areas of the world have been affected by the increase in heat waves, the increase in the magnitude of periods of drought, the decrease and absence of rainfall, and soil degradation [19–21]. All these phenomena are not direct causes of wildfires, but are factors that influence their recurrence and magnitude. Heat, drought, lack of rain, and land degradation affect the availability of fuel material, which favors the spread of wildland fires [4,22]. These factors also affect the intensity of fires and the damage they cause [13,23–25]. An interesting example of the interaction between climate change and wildfires is the increase in ignition points in mountainous areas that are difficult for people to access [26]. Lightning generated in dry electrical storms (without rain) can cause fires in native forests in these areas, where difficult access is an obstacle to their control [27–29].

The relationship between climate change and wildland fires increases concern about its social and environmental impacts. Wildland fires impact people's health due to burns or the large amount of $CO_2$ they generate, produce irreparable damage to infrastructure and homes located in interface areas, and affect productive activities [3,30,31]. On the other hand, wildland fires destroy flora and fauna, generate desertification, and deteriorate biodiversity [20,31]. The interaction with climate change increases the damage capacity of these impacts.

In this scenario, it is necessary to generate tools that allow technical organizations and communities to have information that enables better wildland fire risk management [20,32]. For this, it is essential to advance the study of the behavior of risk factors in particular territorial contexts. This will allow the delivery of valuable and valid information to territorial planners, risk managers, and community leaders about the prevention, preparation, firefighting, and mitigation actions they must promote in their environments. Forest fire risk analysis is, therefore, contextual, since it depends on the characteristics of the socio-ecological systems in which it is carried out. We understand socio-ecological systems as those that emerge from the relationship between biophysical and social factors, sustaining a set of human needs and environmental conditions in interaction [33–36].

From disaster risk management, the concept of risk integrates the following two main components: damage and future projection. Risk has been defined as the probability of occurrence of a future event that can cause possible damage [37,38]. Therefore, the risk of wildland fires refers to the negative impacts that these can have on communities and ecological environments [20]. The probability of wildland fire occurrence depends on the interaction between different geographic, topographic, climatic, land cover, and human action variables [11,39–41]. The interaction between these factors determines the magnitude of the projected damages [11,42].

The social theory of risk expands the analysis of this concept and its application in disaster risk management. This integrates the decision as a relevant component of the risk. The probability of damage is the consequence of an unwanted decision that wants to be avoided. For decisions to manage the risk, it is necessary to know what and how this damage is produced [43]. In the context of wildland fire risk, this means integrating knowledge about what makes wildland fires so that the decisions adopted for their management are those that prevent their occurrence or minimize their damage. In this sense, risk analysis implies anticipating possible negative results of a decision [44,45].

The study of fire risk can be divided between research on propagation and the ignition of forest fires. This work addresses the risk of wildland fire ignition due to its relevance in developing prevention and preparedness strategies. A fire occurs when an ignition source (human or natural) meets the available combustible material [3]. Ignition risk is the probability that a fire will start at a given point in the territory [46]. Depending on the conditions and characteristics of the space where the ignition point is located, the flames can increase in intensity and propagate, developing into a forest fire [5,47]. Ignition depends on a wide range of variables associated with the point where it occurs. These can be

classified into (a) natural conditions, such as plant species and plantations height, humidity, temperature, topography, and local climate, [3,4,11,47,48] and (b) anthropogenic conditions, such as land use and cover, distance from roads, distance from urban or inhabited areas and urban infrastructure [6,20,23,26,47,49]. The ignition risk increases when the associated variables interact at a certain point.

Wildland urban interfaces (WUI) are areas in which these factors interact and are enhanced. Research indicates that the risk of wildland fires is greater when human settlements mix with vegetation [40,50]. This acquires relevance in the context of megafires. The most significant damage caused by wildland fires is in the ecological environment in which they occur. The damage on urban surfaces is usually negligible if we compare it with vegetated or forested surfaces. In megafires, however, the risk of damage in inhabited urban areas increases [51–53]. Their high intensity characterizes megafires compared to the general pattern of wildland fires. Other characteristics are their broad ecological and social impact, the obstacles they impose on their management, and the danger of reactivation [19,54]. All these factors imply that these events tend to cross WUIs, affecting people and urban infrastructure more significantly than forest fires. Along with this, a megafire can change the land cover, reducing the ecosystem services that the land provides to the surrounding populated areas [51,55].

In this study, we analyze the risk of wildland fire ignition by modeling and mapping these events. Our case study is the Maule region in Chile, which was affected by the megafire of 2017. Considering the high vulnerability that this area presents to the impacts of climate change and the uncertainty about its future effects, we developed a model of ignition risk from historical fire data in the region. First, we identify the variables that best explain ignition. Based on these variable values, the model assigned a probability of fire occurrence to points on the map. We then validated the model with data from fires during the megafire and later years. Finally, we identify four interface areas affected by the megafire of 2017 and compare the state of the territory before and after this event. This last exercise aims to analyze the current fire risk conditions in a region that, due to its high exposure to climate change, may once again be affected by events of significant magnitudes, such as the 2017 megafire.

## 2. Case Background

Between January and February 2017, the south-central zone of Chile experienced one of the largest megafires in history [56]. The amount of heat energy released during the months that this event occurred exceeded the scales used internationally until then [57]. Official figures indicate that the fire destroyed 529,974 hectares. Although the greatest damage occurred in forest areas, a set of WUIs was affected, with 3000 homes lost and 11 deceased persons [16,51].

Megafires are usually generated in Mediterranean landscapes, such as the affected area in Chile. The climatic conditions and homogeneous land cover, given the vast extension of forest plantations, were favorable conditions for this event [58,59]. These conditions are characteristic of the Maule region, one of the most affected by the 2017 event. In this region, the fire destroyed 252,556 hectares, equivalent to a third of the forest area of the region [51]. The mega-drought that has affected this region since 2010 led to the fire spread and a high level of damage [57,60,61].

Due to this disaster, the resources provided for firefighting were increased in the country, which positively increased the response capacity of specialized agencies [61]. Little has been addressed, however, in the prevention of wildland fires in science and public policy. We believe studying the risk of wildland fire ignition will contribute to the characterization of areas highly exposed to these fires, information that can be used in prevention policies and actions. On the other hand, the set of conditions that increased the chance of the development of the megafire in the Maule region has not changed in recent years, making it necessary to analyze the risk of ignition of wildland fires and the associated factors.

### 2.1. Study Area
Region of Maule

The study area is located in the Maule Region, in central Chile, between 34°41′ and 36°33′ south latitude. The surface area is 30,296 km$^2$, equivalent to 10% of the national territory [62]. Its population is 1,044,950 inhabitants, with a density of 34.5 inhabitants per square km, with 73% of the population living in urban areas [63]. Its topography integrates mountains of 4000 m.a.s.l. in the Andes Mountains, an intermediate depression, and the Coastal Mountains with mountain ranges with moderate to steep slopes. The climate is a temperate Mediterranean climate, with a dry season of six months in the north and four months in the south. The primary use of the land is grassland and scrubland (25%), agricultural land (22%), and forest plantations (20%) [64]. It has an area of native forest of 581,515 hectares and 634,893 forest hectares [65]. Given the topographical and climatic characteristics, the forest crops are mainly located in the Cordillera de la Costa [64]. For the analysis of the changes in the landscape after the 2017 megafire, we selected the following three communes in the region: Constitución, Empedrado, and Cauquenes (Figure 1). The selected communes correspond to the populations most affected by the 2017 fires [66].

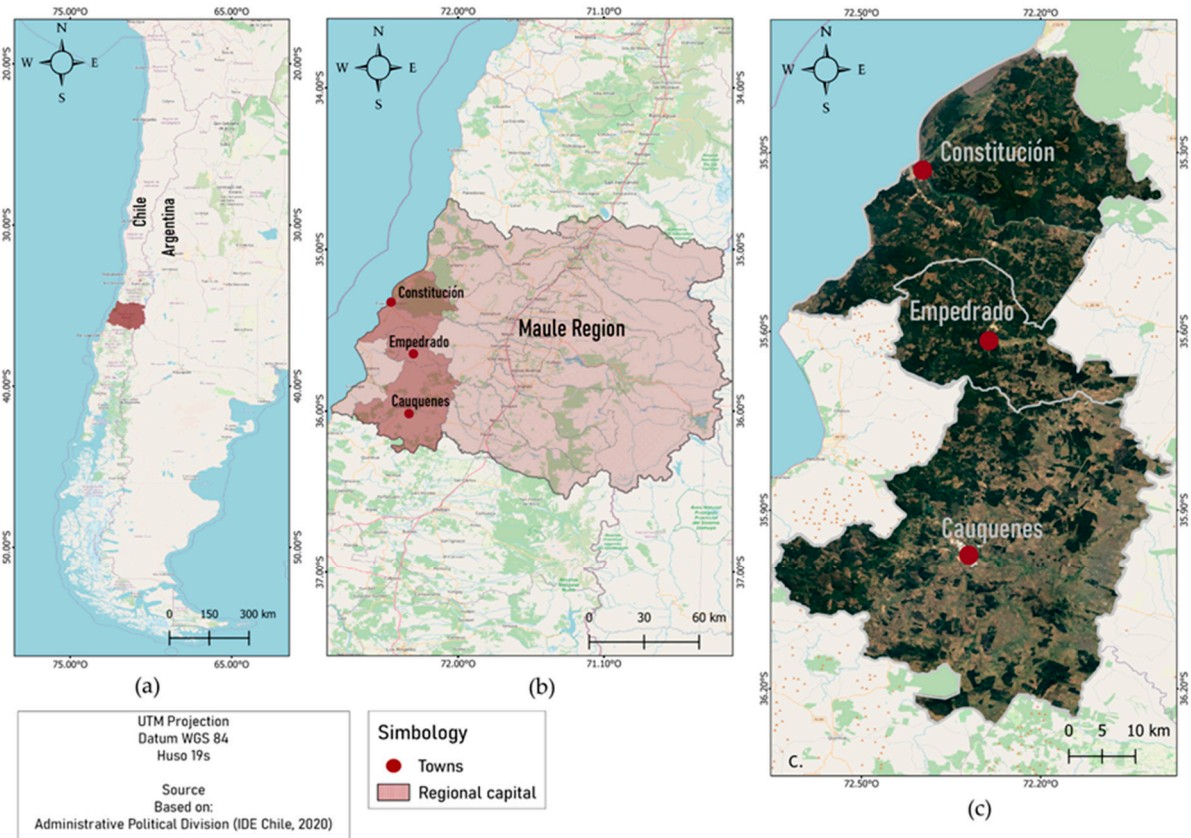

**Figure 1.** (**a**) Location of the study area in Chile. (**b**) Location of the study area on a regional scale. (**c**) Communal scale study area.

It should be noted that in Empedrado, the town of Santa Olga was entirely consumed by the fire, becoming an emblematic case of megafires' impact on a population in a WUI [15,67]. The incident negatively impacted the water and electricity supply and generated contamination of water sources for human and animal consumption. In these communes, the labor sources associated with agricultural and livestock production and the forestry industry were also affected [67].

Constitución, located between latitude 35°19′ south and longitude 72°24′ west, is a coastal city with a total area of 1344 km$^2$ [68]. It is located on the south bank of the mouth of the Maule River in the Pacific Ocean. The maritime influence keeps its daily temperatures

moderate, with an annual average of around 14 °C. It is characterized by hot summers and mild winters, with an average annual rainfall of 662 mm. Its population is 46,068, of which 19% reside in rural areas [63]. The main economic activity is forestry and agriculture. The use of the land is native forest (5%), plantations (41%), thickets (34%), crops, and grasslands (19%) [68].

In the south of the region is the commune of Empedrado, between the coordinates 35°36′ south latitude and 72°16′ west. Its surface is 565 km² and is located in the coastal mountain range. Its annual temperatures range between 10 °C and 33 °C [69]. Its total population is 4142 inhabitants, of which 27% live in rural areas [63]. The main economic activity is agricultural and forestry production, including agriculture, livestock, dairy production, wines, and liquors. Land use is divided into the native forest (17%), plantations (17%), bushes (17%), agricultural (17%), grasslands (17%), meadows (17%), grassland crop rotation (17%) and others (17%) [69].

Between 35°58′ south latitude and 72°18′ west longitude is the commune of Cauquenes, with an area of 2216 km² [70]. Its population is 40,441 inhabitants, with a rural population of 18% [63]. It has a main body of water, the Cauquenes River, whose flow has decreased due to the drought affecting the region. Its climate is a Mediterranean climate, with average temperatures of around 25 °C in January and 7 °C in July. The economy of the commune is diverse and includes manufacturing activities (20.5%), forestry (17.3%), and electricity production (13.2%), among others. Land use is divided into the native forest (32%), forest plantations (32%), and thickets (31%), which together cover 95% of the surface [70].

## 3. Materials and Methods

We developed a model of the risk of ignition of forest fires in the Maule region through a machine learning model. For this, we defined the ignition of forest fires as a dependent variable and selected a set of independent variables as possible predictors of ignition. The construction of the database was in two stages. First, spatially represented points of the ignition variable (binary variable) were generated. Then, the values of the set of independent variables were estimated for each of these points.

The dependent variable was generated from the Corporation Nacional Forestal (CONAF) data, published as official data on its website (CONAF https://www.conaf.cl/incendios-forestales/incendios-forestales-en-chile/estadisticas-historicas/ (accessed on 20 May 2022)). From this, we obtained information on the coordinates, start date, control, cause, and magnitude, of the fires that occurred in the Maule region between 2013 and 2015. The period defined for collecting information was due to the need to have a rich source of available data on fire ignition and independent variables before the megafire of the 2016–2017 season. On the other hand, we decided to generate the model with data before the megafire to contrast it with what happened during that event to validate its results. With this, we looked for a model that allowed us to explain the ex-post distribution of the ignition points of forest fires, while predicting the risk of ignition in the future.

We assigned the value 1 (one) to the points where a fire occurred in the indicated years and a value 0 (zero) to the points where there was no ignition. The assigning points that represented areas with no ignition of wildland fires were carried out randomly. Points with the value 0 were assigned to located areas more than 500 m from an ignition point. For the construction of georeferenced data, we used the geographic information system QGIS (version 3.20.3. for Windows/Copyright © 2000, 2001, 2002, 2007, 2008 Free Software Foundation, Inc. <http://fsf.org/>).

We obtained 3784 points, of which 1892 indicate the ignition of forest fires between 2013 and 2015 in the Maule region; these correspond to the red points in Figure 2. The 1892 blue points generated by the model represent areas with no ignition in the same period.

The independent variables selected for the construction of the model were based on the work of Miranda et al. [40]. According to the authors, the scientific evidence indicates that this set of variables is the one that best represents the territorial characteristics of those places where wildland fires occur in interface areas. Based on the results of this group

of researchers, we worked with 14 independent variables organized into the following 3 classes: (a) human activity, (b) geography and topography, and (c) land cover (Table 1). Each variable was spatially represented in a 30-m resolution raster. To assign the values of the independent variables at each point, we defined a zone of influence of 500 m (centered at the point of ignition). These buffers were built by taking a circumference of a radius of 500 m around each point of the raster. Each one was assigned the value (percentages, averages) of each explanatory variable to avoid bias.

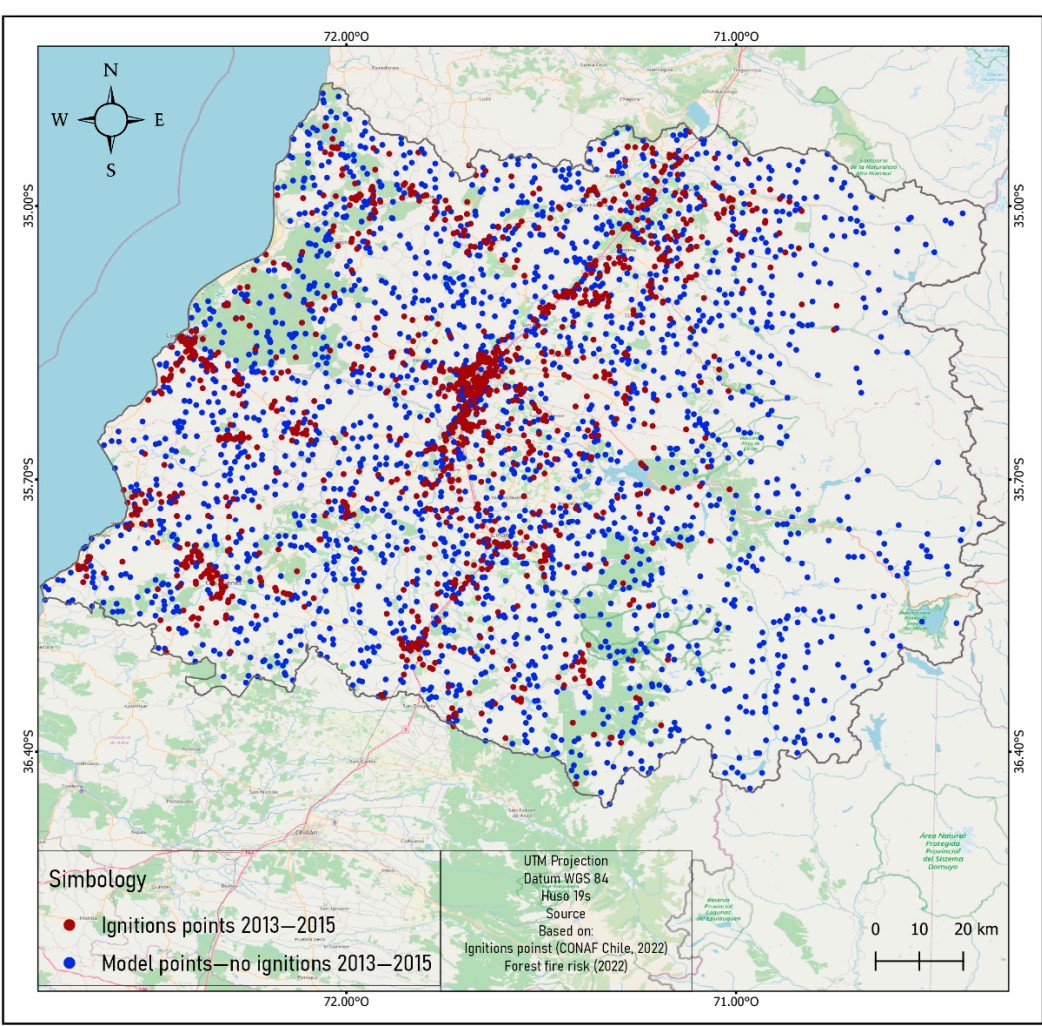

**Figure 2.** Binary dependent variable, ignition of wildland fires in the Maule region between the years 2013 to 2015.

Each of the 3784 points was assigned values for these 14 variables. Thus, the model was trained with the values obtained between 2013 and 2015.

We seek to identify the variables that most influenced the fire's start in the past. With this, we estimate what factors generate a greater probability of a fire. Based on the methodology used by Miranda [40], we use the bagged decision tree, BDT model [75]. This ensemble machine learning method combines different "weak" classification sub-models to obtain a "strong" one. The processing of the model was carried out using the MATLAB R2020a (see: https://www.mathworks.com/products/matlab.html?s_tid=hp_products_matlab (accessed on 10 April 2022)).

Bagging (which is short for bootstrap aggregating) consists of building different submodels using random samples, with replacement, and then assembling the results. Various subsets of the training set data are created. The model has that name, since it trains the submodels using bagging. A model is trained with each subset, and the final

predictions are averaged, making it more robust. An example of a bagging model is shown in Figure 3.

**Table 1.** Independent variables, names, and sources.

| Variables Classes | Name | Label | Source |
|---|---|---|---|
| Human activity | DensPop | Population density of the point (inhabitants/census district) | Instituto Nacional de Estadísticas: https://www.ine.cl/herramientas/portal-de-mapas/geodatos-abiertos (accessed on 2 April 2022) [71] |
| | DistCit | Distance from point to nearest city (m) | |
| | DistRoad_buff | Average distance from the 500 m radius buffer to the nearest road (m) | Ministerio de Obras Públicas: https://ide.mop.gob.cl/geomop/ (accessed on 2 April 2022) [72] |
| Geography and topography | Exposition | Point exposition (indices) | Earth Resources Observation and Science Center (EROS), https://www.usgs.gov/centers/eros/data-tools (accessed on April 2022) [73] |
| | Slope | Point slope (grades) | |
| | Elev | Point elevation (m.a.s.l.) | |
| Land cover | Crop_buff | Proportion of crops in a 500 m radius buffer (proportion) | Zhao Y et al. 2016 [74] |
| | Nat_buff | Proportion of native forest in a 500 m radius buffer (proportion) | |
| | ForPlan_buff | Proportion of forests plantation in a 500 m radius buffer (proportion) | |
| | Grass_buff | Proportion of grassland in a 500 m radius buffer (proportion) | |
| | Scrub_buff | Proportion of scrubs in a 500 m radius buffer (proportion) | |
| | Imper_buff | Proportion of impermeable land in a 500 m radius buffer (proportion) | |
| | BareSoil_buff | Proportion of bare soil in a 500 m radius buffer (proportion) | |
| | LC | Type of land cover predominant in the point (categorical) | |

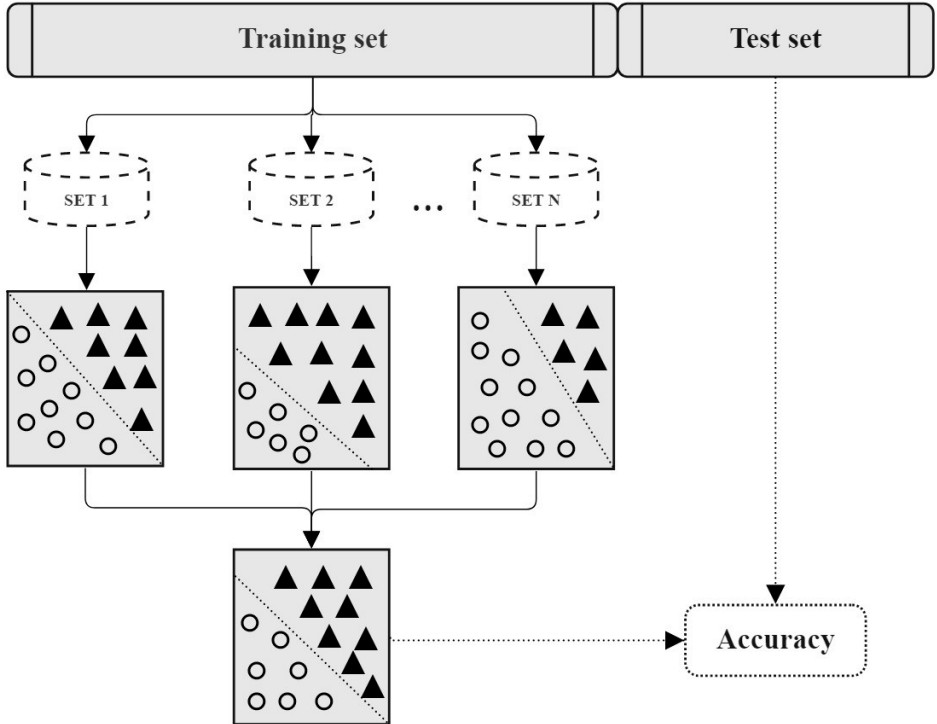

**Figure 3.** Diagram of the bagged model.

For this work, we use the bagged decision tree, where each model is a decision tree. The training set was built with 80% of the database, and the remaining 20% was used to test and verify the accuracy of the results. Once the model was trained, it generated the fire ignition susceptibility classification for each point on the 30 m × 30 m quadrant map.

Finally, a comparative temporal matrix of four WUI was built to analyze changes in the distribution of wildland fire ignition. The matrix was elaborated through the QGIS software. The satellite images come from the Sentinel-2 mission. These images present an atmospheric correction at ground level, providing spectral radiance levels similar to reality. The vector layer of the fire scar was extracted from the Landscape Fire Scars database for Chile [76]. Previous layers corresponding to the ignition points of the 2016–2017 and 2019–2020 seasons were used on the satellite images. It should be noted that the 2020–2021 season was not considered in the analysis. We decided to exclude this season, since the lower number of wildland fires it registers is associated with fewer people in transit in the interface areas, due to the confinement measures adopted in the context of the COVID-19 pandemic.

## 4. Results

### 4.1. Model Fit

The performance of the model was estimated through the global adjustment indicator AUC. This corresponds to the area under the curve ROC (relative operating characteristic), which represents the ratio between the true positives (TPR, true positive ratio) and the false positives (FPR, false positive ratio), as the model predicts values (Figure 4). When the AUC takes a value of 1, the model has perfect prediction. When it takes a value of 0.5, it is a model without explanatory power that does not discriminate between categories. For the model, an AUC value of 0.85 is obtained, which means that it has a high level of fit.

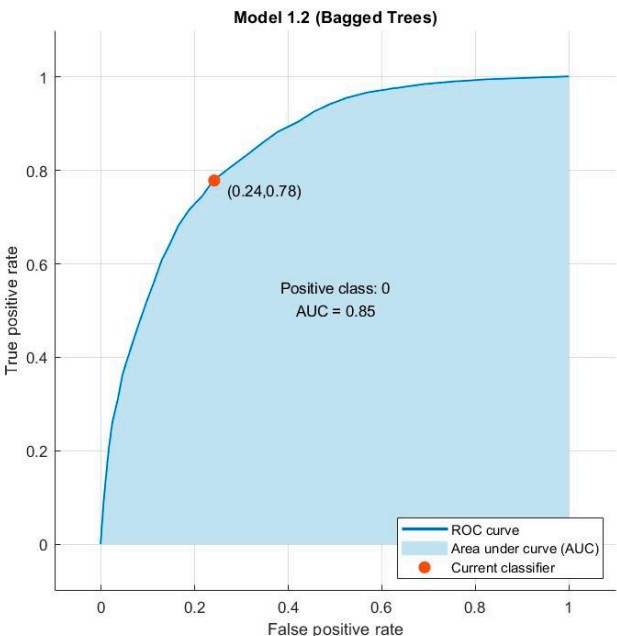

**Figure 4.** ROC curve of the fit of the ignition model.

Another measure of the model's effectiveness is the confusion matrix, which indicates the level of correct and incorrect classifications. In Figure 5, we can observe that 75.8% of values 1 (ignition) and 77.9% of values 0 (not ignition) were correctly classified.

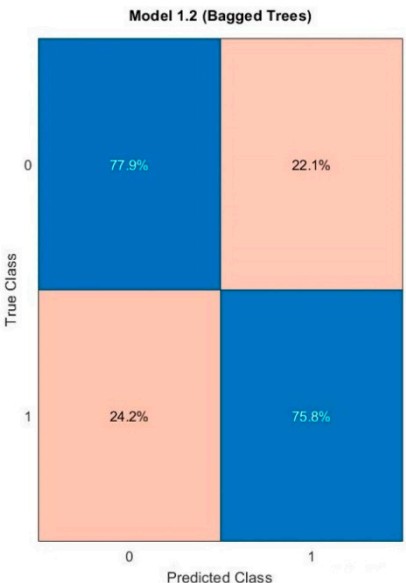

**Figure 5.** Confusion matrix.

One of the advantages of decision tree models is that their results show the explicative importance of the variables. This importance represents the prediction model's error increase after the variable's value has been permuted (separated in a branch of the tree). This is the normalized average of how much this variable changes the final classification result. In this model, the importance of the variables is as follows, as we can observe in Table 2.

**Table 2.** Variables' importance.

| Variable Importance | Variable Label | Variable Name |
|---|---|---|
| 18.9599929 | Proportion of crops in a 500 m radius buffer (proportion) | Crop_buff |
| 15.2302175 | Average distance from the 500 m radius buffer to the nearest road (m) | DistRoad_buff |
| 15.046572 | Distance from point to the nearest city (m) | DistCit |
| 8.80795483 | Proportion of forest plantation in a 500 m radius buffer (proportion) | ForPlan_buff |
| 7.63480897 | Proportion of grassland in a 500 m radius buffer (proportion) | Grass_buff |
| 6.82820899 | Proportion of scrubs in a 500 m radius buffer (proportion) | Scrub_buff |
| 6.35721698 | Point exposition (indices) | Exposition |
| 5.09869124 | Population density of the point (inhabitants/census district) | DensPop |
| 4.0326676 | Point elevation (m.a.s.l.) | Elev |
| 4.01163185 | Point slope (grades) | Slope |
| 2.78965119 | Proportion of impermeable land in a 500 m radius buffer (proportion) | Imper_buff |
| 2.31063774 | Proportion of native forest in a 500 m radius buffer (proportion) | Nat_buff |
| 1.84278487 | Type of land cover predominant in the point (categorical) | LC |
| 1.04896344 | Proportion of bare soil in a 500 m radius buffer (proportion) | BareSoil_buff |

In order of importance, the variables that best explain the ignition of wildland fire in the Maule region are the proportion of crops, the distance to the nearest road, and the distance to the nearest city. The second group of variables with a medium explanatory capacity integrates the proportion of forest plantations, the proportion of grassland, the proportion of scrub, exposure, and population density.

### 4.2. Ignition Risk Model Map

One of the model results is the assignment of ignition risk levels to the different zones of the study area map. This is achieved due to the georeferencing of each point. Each pixel is classified according to its level of susceptibility to the ignition of a forest fire.

In the map of the Maule region, we classify 33,696,273 pixels of a 30 m resolution raster. Each of these pixels is assigned the values of the independent variables of the trained model. The result of the classification delivers values between 0 and 1. It is a continuous variable that represents levels of probability of ignition of forest fires. This graphic representation is built on the map of the region. Figure 6 shows the result of the classification; the blue zones have a lower susceptibility to fire ignition, the yellow zones have a medium probability, and the red zones have a high probability.

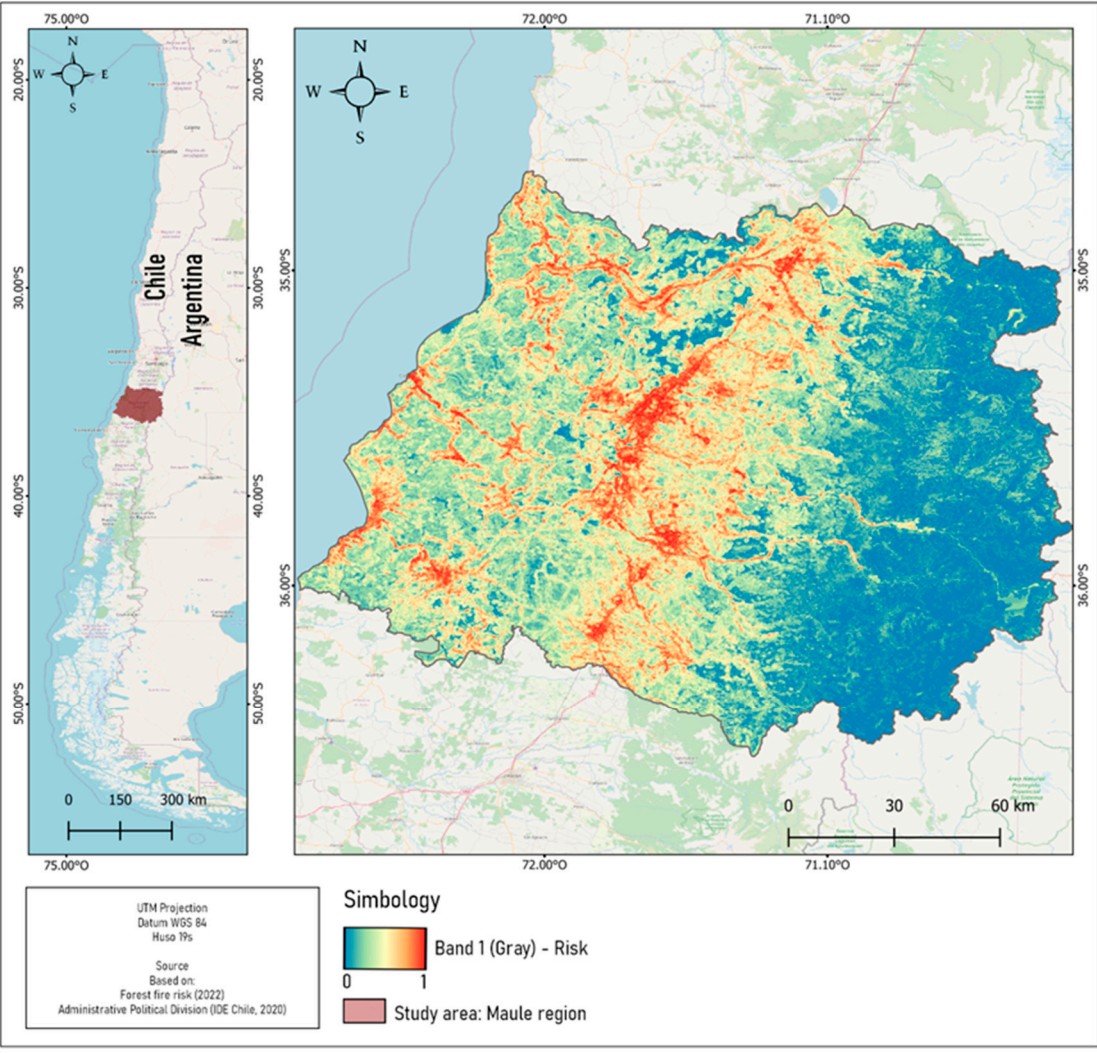

**Figure 6.** Ignition risk map in the Maule region, Chile.

### 4.3. Model Validation

To verify the explanatory capacity of the model, we superimposed on the map that shows the results of the ignition risk model the ignition points recorded in the 2016–2017

season. These account for the distribution of the ignition points of the megafire in the Maule region (red points in Figure 7). In this comparison, it is important to consider that the modeling of the ignition risk was carried out based on data from 2013 and 2015. Figure 7 clearly shows that most of the points where a fire started are in areas the model classified as having a high probability of ignition. These results contribute to validating the model and provide useful information to risk management in the future.

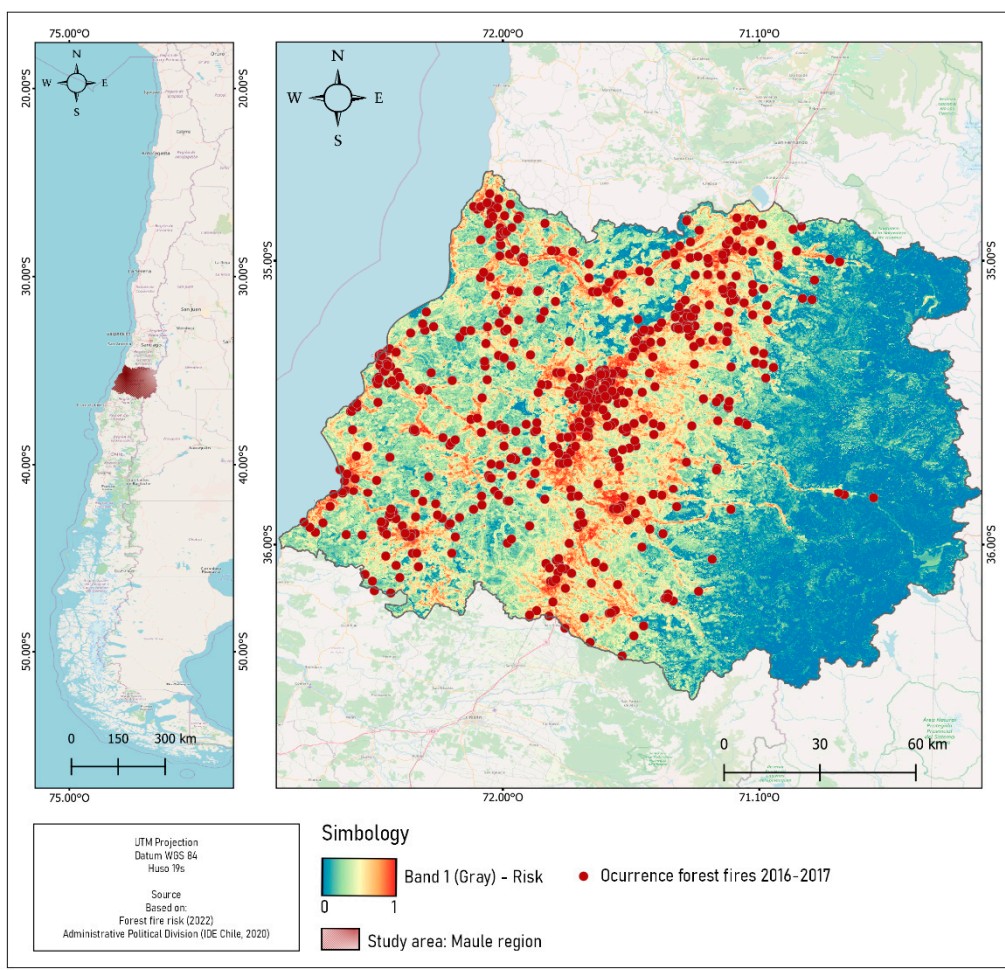

**Figure 7.** Ignition risk map and ignition points during 2016–2017 season.

In addition, we compare the model results with the real ignitions between 2016 and 2020. Table 3 shows that 85% of fires in that period occurred in points classified as medium, high, and very high risk by the model.

**Table 3.** Percentage of fire ignition during 2016–2020 according to model probability levels.

| Ignition Probability | Risk Level | Points Frequency | Ignitions 2016–2020 (%) |
|---|---|---|---|
| 0–20 | Very low | 113 | 3.19% |
| 20–40 | Low | 424 | 11.99% |
| 40–60 | Medium | 728 | 20.58% |
| 60–80 | High | 941 | 26.60% |
| 80–100 | Very high | 1331 | 37.63% |
| | TOTAL | 3537 | 100.00% |

### 4.4. Temporal Comparative Matrix of Interface Zones

To deepen the analysis of our model and generate guidance for decision-makers, professionals in charge of forest fire prevention and preparedness, and communities, we analyzed three interface zones severely damaged by the 2017 megafire (Figure 8). The matrix shows the results of the ignition risk modeling (a), the ignition points of the 2016–2017 season (b), the scar of the 2017 megafire (c), and the ignition points of the 2019–2020 season (d).

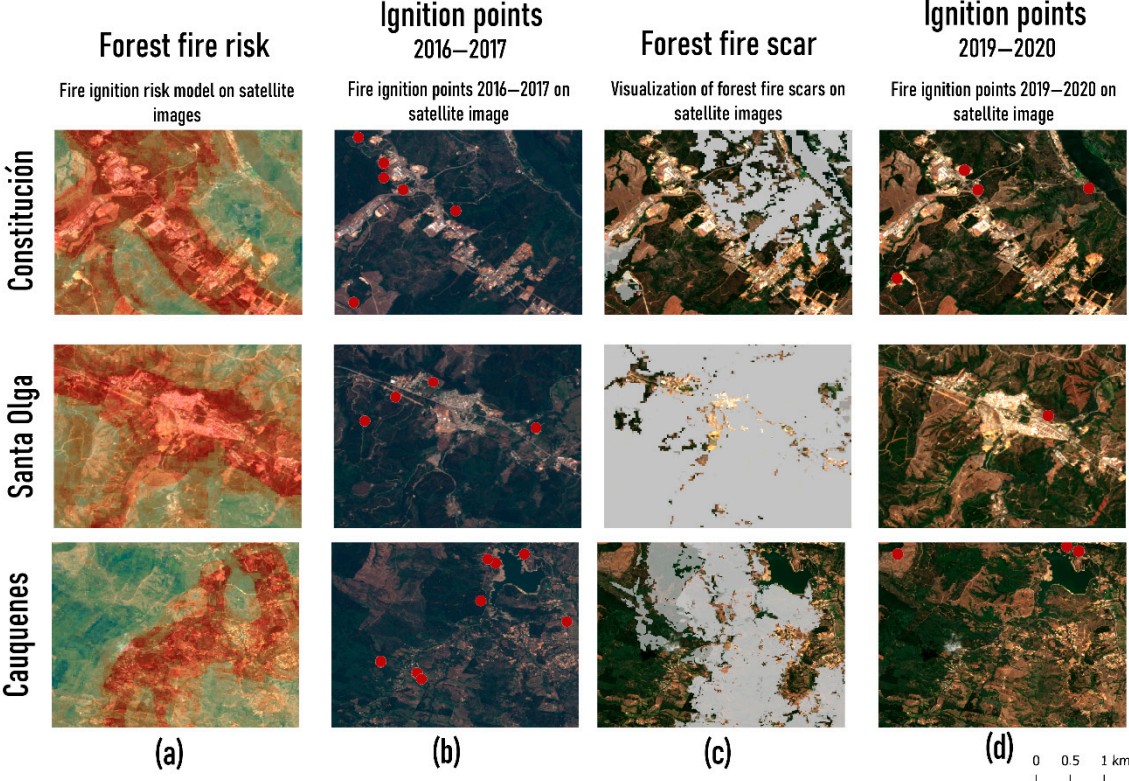

**Figure 8.** Matrix comparative of interface zones.

The results show that fire ignition in the two seasons under analysis occurred in areas the model classifies with a high probability (see columns a, b, and d). In these areas, crop areas of diverse composition can be observed—especially forestry—near the ignition points by roads and cities. These characteristics correspond to the variables with the most significant weight in the probability of ignition of our model.

Comparative analysis of satellite images indicates that the number of ignitions in each WUI is lower in the 2019–2020 season. Changes in the territory's characteristics can also be observed, including a notable decrease in the density of forest crops in each of the analyzed areas (column d). This change in the territory's composition can be attributed to the 2017 megafire scar (column c). A large part of the forest plantations, grassland, and scrubs in these areas was destroyed by the fires of 2016–2017. Our model classifies these variables with a medium influence on the probability of ignition.

It should be noted that the town of Santa Olga was destroyed by the fires of 2017. As a reparative measure, it was quickly rebuilt in the following years. The reconstruction of the town meant an improvement in the conditions and quality of life of its population, which received better quality housing and urban infrastructure that did not exist before the megafire [67]. These improvements, however, do not mean a reduction in the ignition risk in the area adjacent to this town, since it continues to be surrounded by extensive and dense forest plantations.

## 5. Discussion and Conclusions

The ignition risk model generated in this work makes it possible to identify those sectors of the wildland urban interface zones with a higher probability of forest fire ignition. In turn, it tells us what characteristics of the study area make it more prone to ignition. Depending on their predictive capacity, two groups of variables allow us to understand why ignition is more probable in these sectors. The group of variables with the most significant predictive capacity includes the crop proportion, the distance to the nearest road, and the distance to the nearest city. These are variables of anthropogenic origin if we consider that they are associated with productive activities (crops) and urban characteristics (roads and cities). The second group of variables has a medium ability to predict ignition. Among these, we find variables of anthropogenic origin, such as the proportion of forest plantations and population density, and variables of natural origin, such as grasslands, scrubs, and exposure.

The results coincide with previous studies that have verified the coincidence between forest fires' risk and high proportions of land use destined for crops [26]. In this regard, particular importance has been assigned to forest crops in the ignition of fires [22]. It has been shown that the uncontrolled growth of forest species increases the combustible material and the ignition risk [16,22,77–79]. On the other hand, it has been pointed out that replacing the native forest with homogeneous pine and eucalyptus plantations in the central-southern zone of Chile has led to a greater production of large-scale wildland fires [56]. Our results help confirm the strong influence of human activities on forest fire ignition [4,10,20,24,40]. As in our work, the proximity of ignition points to roads and human settlements are factors that previous studies highlight as risk factors [3].

The analysis of the WUI satellite images indicates that the territory's characteristics changed due to the scar from the 2017 megafire. The density of forest plantations is the most evident change. The images show that forest plantations are in the process of recovery. On the other hand, the urban infrastructure has recovered rapidly. This maintains the high levels of risk associated with the distance from roads and cities and population density. In particular, Santa Olga town reconstruction shows that the repair process focuses on restoring the urban infrastructure and improving the living conditions of the people affected by the fires. However, integrating the ignition risk prevalent in the Maule region into decisions is not evident.

It is feasible to interpret that the lower number of ignitions observed in the 2019–2020 season is a consequence of the megafire scar. Variables associated with crops, scrubs, and grasslands decrease their influence on ignition, given that such combustible materials are less dense. However, it is crucial to remember that the risk of ignition will grow as forest crops recover. This information must be incorporated into the decisions adopted for the prevention and preparation of forest fires. Along with this, the authorities and communities must integrate into their decisions how climate change contributes to ignition probability.

The decrease in rainfall, soil degradation, and heat waves are impacts of climate change related to the variables that the ignition risk model classifies with a strong and medium influence. Previous studies have confirmed that these impacts of climate change affect soil humidity, creating conditions for high flammability of combustible materials [4,10,11,56]. The exposure of the land to sunlight is another climate change variable that our model associates with the ignition risk. According to Maniatis (2021), in areas more exposed to sunlight, the vegetation loses humidity and becomes more flammable. The Maule region has been highly impacted by climate change. This condition is not expected to vary, at least in the short and medium term [60,61,80]. For this reason, it is imperative to consider the variables above in wildland fire risk management, especially given the uncertainty that a new megafire might occur in the region.

The creation of models that make it possible to understand and make visible the risk of wildland fire ignition is necessary but not sufficient. If disaster risk is still understood only as the future projection of damage, these models only serve to identify the risk, not to manage it. Forest fire management requires integrating risk as a decision. This means

improving the policies that regulate the density of forest plantations, the maintenance and expansion of firebreaks in interface zones, land use, and post-fire cleaning of areas where tree species' seeds are irregularly disseminated. It also means integrating the ignition risk into the measures to repair the damage caused by wildfires.

Developing ignition models, such as the one in this work, is undoubtedly necessary to manage wildland fires. An example is the verification of sectors with a greater probability of ignition that, according to the model, are strongly associated with the areas where multiple fires occurred in 2017. This information will allow fire governance actors to implement better strategies to prevent wildland fire ignition. Nevertheless, we must be aware that, even so, forest fires will continue to occur. The risk of ignition is higher in regions such as the study area, where climate change has strongly impacted territory characteristics and conditions. For this reason, integrating the ignition risk into decisions also means implementing campaigns and actions to improve the ability to react to this kind of phenomenon.

It should be noted that the model developed in this work accounts for the particularities of the study area. The model was trained with historical data from the Maule region in Chile. Therefore, the results obtained are valid only for the case analyzed. However, working with the bagged decision tree model has the following significant advantage: it is possible to replicate it in other areas with different characteristics. It is possible because the model learns from the data provided to it. In addition, variables such as human activity, geography, topography, and land cover can be obtained for other areas of interest. The only limitation is that these layers of information, particularly land cover, are scarce in many countries.

One of the challenges that arise from this work is to continue deepening the characterization of the ignition risk. We believe that obtaining better views of the interface areas and the factors associated with risk is necessary. This will allow professionals and people to integrate this information. Visually identifying the components and characteristics of the territories where the ignition risk is higher can contribute to a better understanding of how to prevent and prepare for wildland fires.

**Author Contributions:** Conceptualization, G.A.d.l.C.; methodology, G.A.d.l.C., G.A., R.C. and C.A.; software, G.A., R.C. and C.A.; validation, G.A and R.C.; formal analysis, G.A.d.l.C., G.A. and R.C; investigation, G.A.d.l.C., C.A. and P.O.; resources, G.A.d.l.C., G.A., C.A. and P.O.; data curation, G.A and C.A.; writing—original draft preparation, G.A.d.l.C.; writing—review and editing, G.A.d.l.C.; visualization, C.A.; supervision, G.A.d.l.C.; project administration, G.A.d.l.C.; funding acquisition, G.A.d.l.C. All authors have read and agreed to the published version of the manuscript.

**Funding:** This research was funded by the National Agency of Research and Development of Chile, ANID, Fondecyt N° 11190483.

**Institutional Review Board Statement:** Not applicable.

**Informed Consent Statement:** Not applicable.

**Data Availability Statement:** The dependent variable was generated from the National Forestry Corporation of Chile (CONAF) https://www.conaf.cl/incendios-forestales/incendios-forestales-en-chile/historical-statistics/ (accessed on 2 April 2022)). The independent variables can be found at the Instituto Nacional de Estadísticas, https://www.ine.cl/herramientas/portal-de-mapas/geodatos-abiertos (accessed on 2 April 2022); Ministerio de Obras Públicas, https://ide.mop.gob.cl/geomop/ (accessed on 2 April 2022); Earth Resources Observation and Science Center (EROS), https://www.usgs.gov/centers/eros/data-tools (accessed on 22 May 2022). Sentinel-2 images for mapping-based use are located on the Copernicus Open Access Hub (https://scihub.copernicus.eu/dhus/#/home, accessed 22 May 2022). The vector layers used for the administrative political division can be downloaded from the Geospatial Data Infrastructure (IDE, Chile) (https://www.ide.cl/index.php/limites-y-fronteras/item/1528-division-administrative-policy-2020, accessed on 11 April 2022).

**Conflicts of Interest:** The authors declare no conflict of interest.

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
