# Peer review of "Modeling the Ignition Risk: Analysis before and after Megafire on Maule Region, Chile"

_applsci, doi:10.3390/app12189353_

Round 1
Reviewer 1 Report
Direction of research is very important; this request very high reliability of inferences. Time and space data are always contained noisy statistical uncertainty which may cause essential mistakes in parametric inferences, see monography Igor Zurbenko, Devin Smith, Amy Potrzeba-Macrina, Barry Loneck, Edward Valachovic and Mingzeng Sun, High-Resolution Noisy Signal and Image Processing, Cambridge Scholars Publishing 2021, 375 pages.
https://www.cambridgescholars.com/product/978-1-5275-6293-6
Local spatial and time smoothing must be applied to the data before modelling to avoid erroneous results. Robust against uncertainty results can be well different.
I may advise to represent data (figs.2, 7 ) on a spatial grid of 1km and make spatial smoothing following Ch.2 of referred book. Smooth components can be applicable to regression analysis like authors already were doing. Such an operation is actually very close to authors developments but will provide results without destructive effect of short fluctuations.
Author Response
We greatly appreciate the reviewer's suggestion. After reviewing the suggested monograph, we realize that it is not feasible to do what he asks of us. This is because our model is trained with a database built with fire ignition points and not with complete satellite images. Therefore, the suggested pre-processing cannot be applied to the training data. Since this was not made clear in the text, we improved the explanation of the methodology (See 271-273, 292-295, and 360). Given your concern about our model's reliability, uncertainty, and error, we included a new analysis to reinforce the validation of the results we obtained (382-386).
Reviewer 2 Report
Overall, this was an interesting paper and should be accepted for publication after some revisions and some more analysis needs to be carried out.
There are three areas that need to be improved. The first is the English expression. There were some parts that were very had to read and understand. I will outline some of these.
Page 1
* line 4 - if possible, put a space between "1 2 3" and "*" ie "1 2 3 *" for the affiliations.
* line 14 - change "We used human activity, ..." with "We use factors such as human activity, ..."
* line 15 - replace "cover variables. We work with Bagged ..." with "cover variables to develop a Bagged ..."
* line 16 - be consistent with regards to "megafire". It is sometimes "Megafire", "megafire"
or "mega-fire"
* line 31 - put a space between words and a reference. For example "great global concern[1]." should be "great global concern [1]."
* line 42 - replace "deepen the analysis of" with "analyse"
* line 43 - remove "complexity of the", replace "favor" with "affect" and "fire's" with "fire"
Page 2
* line 44 - replace "Added to this is the need to integrate climate change's impacts on the generation of these socio-environmental disasters into the analysis." with "In addition, there is a need to connect this with the impacts on the socio-environmental disasters."
* line 47 - replace "on" with "in" and "environment. This is given the connection of its impacts with the exposure of socio-ecological systems to disaster risks" with "environment, and in particular,
the disaster risks"
* line 50 - replace "Because of" with "Due to"
Page 3
* line 101 - replace "exploding" with "developing into"
* line 129 - the sentence seems to end prematurely
* line 138 - replace the dashes "-" with commas ","
* line 145 - replace "Because of" with "Due to"
* line 150 - replace "favored the" with "increased the chance of the"
Page 4
* line 157 - replace "Its surface" with "The surface area"
* line 158 - replace "34.5%, 73%" with "34.5 inhabitants per square km, with 73%"
* line 170 - replace "flames" with "fire"
* line 178 - to be consistent with later in the text, replace "13.8oC" with "around 14oC"
Page 5
* line 240 - put in the full name "Corporacian Nacional Forestal (CONAF)"
* line 241 - check the link. I got a "404 - page not found" error
Page 6
* line 258 - the points look gray, not blue to me
Page 7
* line 274 - remove "Through a Machine Learning model"
* line 279 - replace "the MATLAB R2020a software" with "MATLAB R2020a" and include a reference to MathWorks
Page 9
* line 328 - it is probably best to say "the fire was correctly predicted (1) or not (0)", rather than 1 and 0
* Figure 5 - this could be simplified greatly. Firstly, the right hand table is not needed. Secondly, this could be reduced down to simple table
Predicted class
0 1
True class 0 78% 22%
1 24% 76%
Page 10
* Figure 6 - use the labels (eg Proportions of crops in a 500 m radius of buffer) from Table 1, not the name (eg Crop_buff)
- What is the scale of the x-axis? I assume it is a percentage of the model significance? However, this is probably better conveyed as a table, not a figure.
Page 11
* line 368 - replace "figure 8" with "Figure 8"
Page 12
* line 379 - the figure number is missing
Even though the model presented does give reasonable good accuracy, it is a complicated model with 13 factors. I think it would be a worthwhile exercise to also develop a simpler model using only the top 3 or 4 factors.
Author Response
Thank you very much for your thorough review. We applied all the corrections as suggested. Along with this, we check the grammar and spelling of the document using specialized software.
Below we respond to other observations made:
Observation: line 241 - check the link. I got a "404 - page not found" error
Response: now the link works correctly
Observation: line 258 - the points look gray, not blue, to me.
Response: we improved the map so that the blue color is seen more clearly.
Observation: Figure 5 - this could be simplified greatly. Firstly, the right hand table is not needed. Secondly, this could be reduced down to simple table
Response: We simplify the figure so that the information provided is more clearly
Observation: Figure 6 - use the labels (eg Proportions of crops in a 500 m radius of buffer) from Table 1, not the name (eg Crop_buff). What is the scale of the x-axis? I assume it is a percentage of the model significance? However, this is probably better conveyed as a table, not a figure.
Response: we replace the figure with a table that shows the name and label of each variable
Observation: Even though the model presented does give reasonable good accuracy, it is a complicated model with 13 factors. I think it would be a worthwhile exercise to also develop a simpler model using only the top 3 or 4 factors.
Response: We appreciate the suggestion. However, we decided to keep the 14 variables we worked on, comparing our results with those obtained by Miranda et al. (2021). This paper is the basis for the choice of variables and the model, so we wanted to be faithful to what was proposed by these authors. We included a new analysis to reinforce the validation of the results we obtained (382-386).
Round 2
Reviewer 1 Report
Smooth ignition maps by spatial filtering are still required. They will be display potential spatial probability in future. It will be essential step for a future prediction of new ignitions. Current maps address just separate points in the past. Smooth maps by statistical software which I recommended before will provide prediction maps.
Author Response
We again appreciate your comments and concerns about the validity of our model. Regarding your observations in the book by Zurbenko et al. (2021) they propose the Kolmogorov-Zurbenko Filter to deal with noise and artifacts in high-resolution time-spatial data. They successfully use this filter on continuous data (i.e., temperature in Chapter 2) and the smoothed version improves the consistency of the original database. However, our study deals with binary data: wildfire ignition points, defined as the point from which the wildfire starts. This is the case for a large corpus of scientific literature on wildfire modeling (Alcasena et al., 2016; Carrasco et al., 2021; Duane et al., 2021; Hysa & BaÅŸkaya, 2019; Miranda et al., 2020; Rodrigues & De la Riva, 2014; Sivrikaya & Küçük, 2022; Tien Bui et al., 2017, 2019). The decision to work with non-smoothed binary data as a dependent variable is because local phenomena drive wildfire ignition (e.g., the presence of roads, the presence of electric substations, or the electric grid, terrain slope, terrain aspect, etc.) and wildfire modeling is focused on finding the importance of certain variables in these sites.
Nonetheless, spatial data's noise and artifacts in independent variables should be dealt with using interpolation or filter techniques. To smooth independent variables, we define a 500-meter buffer (centered in the ignition point) used to estimate explaining variables included in the model (e.g., the proportion of crops or plantation forests). We acknowledge that this explanation was not explicit in our previous manuscript, and we add a new paragraph explaining this procedure in more detail (see lines 270-275).
Even though we think that is unnecessary to smooth wildfire ignition point data - given the abovementioned arguments-, we use a Kolmogorov-Zurbenko filter (KZ) using various windows (m parameter). To do so, we use the R package called ‘kza’, developed by Brian Close, Igor Zurbenko, and Mingzeng Sun (https://cran.r-project.org/web/packages/kza/kza.pdf). The input was a raster (matrix) of ~1km (0.01º) with the sum of ignition points in each cell. We test the KZ filter with 2, 3, 5, and 10 value windows (m parameter) with 10 iterations each. Figure 1 shows that increasing the m parameter produces a broader representation of wildfire ignition points and reduces the range of the output variables (as a result of a broader moving average). Figure 2 shows that the proportion of cells >= 0.5, which we can include as 1 in the Bagged Decision Tree model, is restricted to six small areas of our case study, reducing the variance of the independent variables artificially. Hence, training a Bagged Decision Tree model only with these cells as ignition points could introduce significant biases in the explaining capabilities of the model and our results.
You can access our data and records of the processing carried out in:
https://drive.google.com/file/d/1mNMv3F4SeDS0mMq3CGu-a3p1tSlP3uzV/view?usp=sharing
We have attached for you the figures and cited bibliography.
Best regards
